# ABCDE: Approximating Betweenness-Centrality ranking with progressive-DropEdge

Martin Mirakyan[1,2]

[1] Information Systems and Computer Engineering, Instituto Superior Técnico, Lisbon, Lisboa, Portugal
[2] YerevaNN, Yerevan, Yerevan, Armenia



## ABSTRACT

Betweenness-centrality is a popular measure in network analysis that aims to describe the importance of nodes in a graph. It accounts for the fraction of shortest paths passing through that node and is a key measure in many applications including community detection and network dismantling. The computation of betweenness-centrality for each node in a graph requires an excessive amount of computing power, especially for large graphs. On the other hand, in many applications, the main interest lies in finding the top-k most important nodes in the graph. Therefore, several approximation algorithms were proposed to solve the problem faster. Some recent approaches propose to use shallow graph convolutional networks to approximate the top-k nodes with the highest betweenness-centrality scores. This work presents a deep graph convolutional neural network that outputs a rank score for each node in a given graph. With careful optimization and regularization tricks, including an extended version of DropEdge which is named Progressive-DropEdge, the system achieves better results than the current approaches. Experiments on both real-world and synthetic datasets show that the presented algorithm is an order of magnitude faster in inference and requires several times fewer resources and time to train.

## INTRODUCTION

Conducting network analysis has been a prominent topic in research, with applications spanning from community detection in social networks (*Behera et al., 2020b*, *2016*), to detecting critical nodes (*Behera et al., 2019*), to hidden link prediction (*Liu et al., 2013*). One of the more fundamental metrics for determining the importance of each graph node for network analysis is betweenness-centrality (BC). BC aims to measure the importance of nodes in the graph in terms of connectivity to other nodes *via* the shortest paths (*Mahmoody, Tsourakakis & Upfal, 2016*). It plays a big role in understanding the influence of nodes in a graph and, as an example, can be used to discover an important member, like a famous influencer or the set of the most reputable users in a network (*Behera et al., 2019*).

Corresponding author
Martin Mirakyan,
martin@yerevann.com

Computing the measure can be very resource-demanding especially for the large graphs. The fastest algorithm for computing exact betweenness-centrality in a given graph is the Brandes algorithm (*Brandes, 2001*) which has $\mathcal{O}(|V||E|)$ time complexity for unweighted networks and $\mathcal{O}(|V||E| + |V|^2 \log|V|)$ for weighted ones, where $|V|$ denotes the number of nodes and $|E|$ denotes the number of edges in the graph. As $\mathcal{O}(|V||E|)$ can grow very fast with the increase in the network size, several approximation algorithms based on sampling have been proposed (*Mahmoody, Tsourakakis & Upfal, 2016*; *Riondato & Kornaropoulos, 2014*). However, along with the growth in the size of the graph, we face tantamount increases in the execution time and proportional decreases in the accuracy of the prediction. In many applications, the computation of the betweenness-centrality needs to be fast enough to handle dynamic changes in the graph (*Fan et al., 2019*).

Although several distributed computing algorithms exist for calculating betweenness-centrality (*Naik et al., 2020*; *Behera et al., 2020a*, *2019*), where the authors propose approaches to compute BC of a network using map-reduce in a distributed environment, this work focuses on a single machine algorithm. The model works on a single machine and requires only a single GPU for training and can perform predictions on relatively big networks without the need for many machines or excessive computational power.

In fields such as social network analysis and network dismantling it is at times far more important to compute the relative importance of the nodes in the graph rather than obtain the exact scores of betweenness-centrality (*Holme et al., 2002*). Several recent works like *Fan et al. (2019)* and *Maurya, Liu & Murata (2019)* have proposed to reformulate the problem into a learning-to-rank problem with the aim to learn a function that would map the nodes in the input graph to relative ranking BC scores. So, instead of computing the exact scores, the task is changed into finding the correct order of the nodes with respect to their betweenness-centrality.

Instead of using approximation techniques like sampling, recent approaches have proposed to train a graph convolutional neural network on synthetic small graphs that would learn to rank nodes based on their BC and would be able to generalize on bigger real-world graphs (*Fan et al., 2019*).

In general, it is hard to avoid over-fitting and over-smoothing when training deep graph convolutional neural networks (*Rong et al., 2020*). In order to generalize better (over-fitting) on small datasets and avoid obtaining uninformative representations for each node (over-smoothing) in deep models, *Rong et al. (2020)* proposes to use the DropEdge technique while training the network. During the training procedure, they drop a set of random edges from the graph before feeding it to the model. This work further develops this idea by introducing the Progressive-DropEdge technique, which drops a random set of edges, with diminishing probability, based on the depth of a layer in the model. Introducing Progressive-DropEdge in the training procedure improves the performance of the model, especially on larger real-world networks.

This paper focuses on the benchmark of ranking based on betweenness-centrality proposed by *Fan et al. (2019)* as they include various real-world and synthetic datasets and detailed comparisons with other approximation algorithms. The main contributions are threefold:

- First, Progressive-DropEdge is introduced in the training procedure which acts as regularization and improves the performance on large networks.
- Second, deeper graph convolutional networks are shown to be able to have fewer parameters and be more efficient than more shallow alternatives leading to state-of-the-art results while being by an order of magnitude faster.
- Finally, the presented training procedure converges faster and requires fewer resources which enables training on a single GPU machine.

The approach is named ABCDE: Approximating Betweenness-Centrality ranking with progressive-DropEdge.

The source code is available on GitHub: https://github.com/MartinXPN/abcde. To reproduce the reported results one can run:

```
$ docker run martin97/abcde:latest
```

## RELATED WORK

### Betweenness centrality

The best-known algorithm for computing exact betweenness-centrality values is the Brandes algorithm (*Brandes, 2001*) which has $\mathcal{O}(|V||E|)$ time complexity for unweighted graphs and $\mathcal{O}(|V||E| + |V|^2 \log |V|)$ for weighted ones, where $|V|$ denotes the number of nodes and $|E|$ denotes the number of edges in the graph. To enable approximate BC computation for large graphs several approximation algorithms were proposed which use only a small subset of edges in the graph. *Riondato & Kornaropoulos (2014)* introduce the Vapnik-Chervonenskis (VC) dimension to compute the sample size that would be sufficient to obtain guaranteed approximations for the BC values of each node (*Fan et al., 2019*). If $V_{max}$ denotes the maximum number of nodes on any shortest path, $\lambda$ denotes the maximum additive error that the approximations should match, and $\delta$ is the probability of the guarantees holding, then the number of samples required to compute the BC score would be $\frac{c}{\lambda^2} \left( \lfloor \log(V_{max} - 2) \rfloor + 1 + \log \frac{1}{\delta} \right)$. *Riondato & Upfal (2018)* use adaptive sampling to obtain the same probabilistic guarantee as *Riondato & Kornaropoulos (2014)* with smaller sample sizes. *Borassi & Natale (2019)* propose a balanced bidirectional breadth-first search (BFS) which reduces the time for each sample from $\mathcal{O}(|E|)$ to $\mathcal{O}(|E|^{\frac{1}{2} + \mathcal{O}(1)})$. Yet both approaches require a second run of the algorithm to identify top-k nodes with the highest betweenness-centrality scores.

*Kourtellis et al. (2012)* introduces another metric that is correlated with high betweenness-centrality values and computes that metric instead, to identify nodes with high BC scores. *Borassi & Natale (2019)* propose an efficient way of computing BC for top-k nodes, which allows bigger confidence intervals for nodes with well-separated betweenness-centrality values.

*Fan et al. (2019)* and *Maurya, Liu & Murata (2019)* propose a shallow graph convolutional network approach for approximating the ranking based on the betweenness-centrality of nodes in the graph. They treat the problem as a learning-to-rank problem and approximate the ranking of vertices based on their betweenness-centrality.

## Deep graph convolutional networks

Graph Convolutional Networks (GCNs) have recently gained a lot of attention and have become the *de facto* methods for learning graph representations (*Wu et al., 2019*). They are widely used in many graph representation tasks. Yet, different studies have different findings regarding the expressive power of GCNs as the network depth increases. *Oono & Suzuki (2020)* claims that they do not improve, or sometimes worsen their predictive performance as the number of layers in the network and the non-linearities grow. On the other hand, *Rong et al. (2020)* claims that removing random edges from the graph during training acts as a regularisation for deep GCNs and helps to combat over-fitting (loss of generalization power on small datasets) and over-smoothing (isolation of output representations from the input features with the increase in network depth). They empirically show that this trick, called DropEdge, improves the performance on several both deep and shallow GCNs.

## PRELIMINARIES

Let $G = (V, E)$ denote a network where each node has a representation $X_v \in \mathbb{R}^c$ for $v \in V$, where $c$ denotes the dimensionality of the representation, $d_v$ denotes the degree of the vertex $v$, $|V|$ denotes the number of nodes and $|E|$ denotes the number of edges in the graph.

Betweenness-centrality accounts for the significance of individual nodes based on the fraction of shortest paths that pass through them (*Mahmoody, Tsourakakis & Upfal, 2016*). Normalized betweenness-centrality for node $w$ is defined as:

$$b(w) = \frac{1}{|V|(|V| - 1)} \sum_{u \neq w \neq v} \frac{\sigma_{uv}(w)}{\sigma_{uv}} \qquad (1)$$

where $|V|$ denotes the number of nodes in the network, $\sigma_{uv}$ denotes the number of shortest paths from $u$ to $v$, and $\sigma_{uv}(w)$ the number of shortest paths from $u$ to $v$ that pass through $w$.

## METHOD

### Input features

For the input, the model only needs the structure of the graph $G$ represented as a sparse adjacency matrix, and the degree $d_v$ for each vertex $v \in V$. In comparison to this method, *Fan et al. (2019)* uses two additional features for each vertex, which were calculated based on the neighborhoods with radii of sizes one and two for each node. Yet, in this approach, having only the degree of the vertex and the network structure itself is sufficient to approximate the betweenness-centrality ranking for each node. So, the initial feature vector $X_v \in \mathbb{R}^c$ for vertex $v$ is only a single number—the degree of the vertex, which is enriched in deeper layers of the model.

### Output and loss function

For each node $v$ in the graph $G$, the model predicts the relative BC ranking score, meaning that for each input $X_v$ the model only outputs a single value which represents the predicted ranking score $y_v \in \mathbb{R}$. As the output is the relative ranking score, the loss function is

chosen to be a pairwise ranking loss follow the approach proposed by *Fan et al. (2019)*. To compute the pairwise ranking loss, $5|V|$ node pairs $(i,j)$ are randomly sampled, following (*Fan et al., 2019*) binary cross-entropy between the true order and the predicted order of those pairs is computed. So, having the two ground truth betweenness-centrality values $b_i$ and $b_j$ for $i$ and $j$ pair, and their relative rank $y_i$ and $y_j$, the loss of a single pair would be:

$$C_{i,j} = -\sigma(b_i - b_j) \cdot \log \sigma(y_i - y_j) - (1 - \sigma(b_i - b_j)) \cdot \log(1 - \sigma(y_i - y_j)) \tag{2}$$

where $\sigma$ is the sigmoid function defined as $1 / (1 + e^{-x})$. The total loss would be the sum of cross entropy losses for those pairs:

$$L = \sum_{i,j \in 5|V|} C_{i,j} \tag{3}$$

## Evaluation metrics

As the baseline proposed by *Fan et al. (2019)* is adopted, the evaluation strategy is also the same. There are several metrics presented in the baseline. Kendall tau score is a metric that computes the number of concordant and discordant pairs in two ranking lists and is defined as:

$$K(l_1, l_2) = \frac{2(\alpha - \beta)}{n \cdot (n - 1)} \tag{4}$$

where $l_1$ is the first list, $l_2$ is the second list, $\alpha$ is the number of concordant pairs, $\beta$ is the number of discordant pairs, and $n$ is the total number of elements. The range of the metric is $[-1; 1]$ where 1 means that two ranking lists are in total agreement and $-1$ means that the two lists are in total disagreement.

Top-$k$% accuracy is defined as the percentage of overlap between the top-$k$% nodes in the predictions and the top-$k$% nodes in the ground truth list:

$$\text{Top-}k\% = \frac{\{\text{predicted-top-}k\%\} \cap \{\text{true-top-}k\%\}}{\lceil |V| \times k\% \rceil} \tag{5}$$

In these experiments, top-1%, top-5%, and top-10% accuracies as well as the Kendall tau score are reported.

## Training data

The training data is generated similar to *Fan et al. (2019)*. Random graphs are sampled from the *powerlaw* distribution during training. The exact betweenness-centrality scores are computed for those graphs and are treated as the ground truth. As their sizes are small, the computation of the exact betweenness-centrality score is not computationally demanding. To avoid over-fitting on those graphs they are regenerated every 10 epochs. Each training graph is reused eight times on average during a single training epoch.

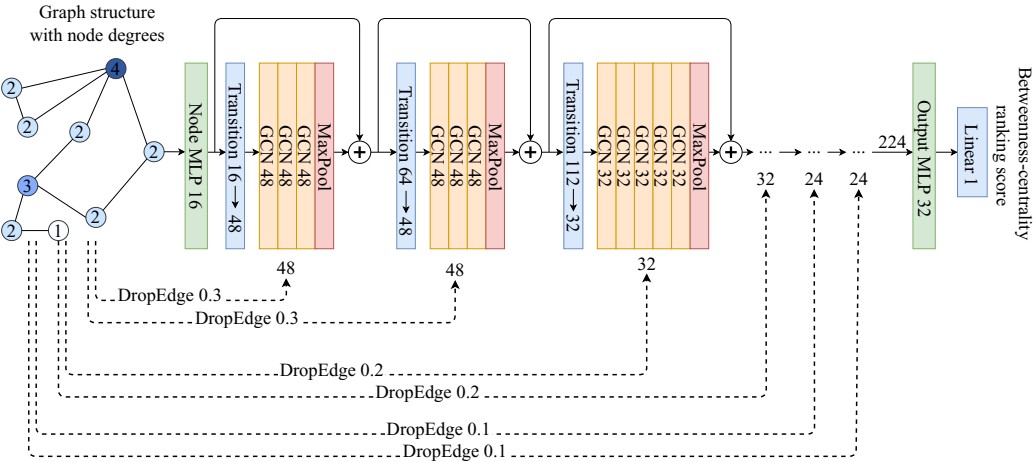

**Figure 1 ABCDE model architecture.** Each Transition block is a set of {Linear → LayerNorm → PRelu → Dropout} layers, while each GCN is a set of {GCNConv → PReLU → LayerNorm → Dropout}. ⊕ symbol is the concatenation operation. Each MaxPooling operation extracts the maximum value from the given GCN block.               

## Model architecture

The model architecture is a deep graph convolutional network which consists of a stack of GCN layers and MaxPooling operations presented in Fig. 1. A GCN operation for a node $v$ which has a neighborhood $N(v)$ is defined as:

$$H_v = W \cdot \sum_{u \in N(v)} \frac{1}{\sqrt{d_v + 1} \cdot \sqrt{d_u + 1}} h_u \tag{6}$$

where $h_u$ is the input vector representation of the node $u$, $d_v$ and $d_u$ are the degrees of the vertices $v$ and $u$ accordingly, $H_v$ is the output vector representation of the node $v$, and $W$ is a learnable matrix of weights.

The model takes the input representation $X_v$ of vertex $v$ and maps it to an intermediate vector representation which is followed by several blocks of GCNs with different feature sizes, followed by MaxPooling operations which reduce the extracted features in the block to a single number for each vertex. Each GCN block is followed by a transition block which is a fully connected single layer that maps the sizes of the previous GCN block to the current one.

For every GCN block, a different amount of random edge drops is applied which is called Progressive-DropEdge. In these experiments the model best scales when the probability of dropping an edge is higher in the initial GCN blocks, while slowly decreasing the probability as the layers approach the output. That helps the model to focus on more details and have a better, fine-grained ranking score prediction. To avoid having isolated nodes only the edges of vertices with degrees higher than 5 are dropped.

## Implementation details

The MLPs and transition blocks follow the {Linear → LayerNorm → PReLU → Dropout} structure, while GCN blocks follow the {GCNConv → PReLU → LayerNorm → Dropout}

structure. The initial MLP that maps the input $X_v$ to an intermediate representation has a size of 16. There are six blocks of GCNs in total. The number of GCNConvs in the blocks are {4, 4, 6, 6, 8, 8}, while their sizes are {48, 48, 32, 32, 24, 24}. The Progressive-DropEdge for each block is applied with probabilities {0.3, 0.3, 0.2, 0.2, 0.1, 0.1}. Gradients are clipped after the value 0.3.

For training and validation, random graphs from the *powerlaw* distribution are sampled using the NetworkX library (*Hagberg, Swart & Chult, 2008*), having nodes from 4,000 to 5,000 with a fixed number of edges to add ($m = 4$), and the probability of creating a triangle after adding an edge ($p = 0.05$) (following *Fan et al. (2019)*). For each training epoch, 160 graphs are sampled, while during validation 240 graphs are used for stability. The batch size is set to 16 graphs per step and the training lasts for at most 50 epochs. The training is stopped whenever Kendall Tau on the validation set does not improve for five consecutive epochs. Adam optimizer (*Kingma & Ba, 2014*) is used with an initial learning rate of 0.01 and the learning rate is divided by 2 if the validation Kendall score does not increase for two consecutive epochs.

The GCN training is implemented in Pytorch (*Paszke et al., 2019*) and Pytorch Geometric (*Fey & Lenssen, 2019*) libraries. All the weights are initialized with their default initializers. The ground truth betweenness-centrality values for training graphs are calculated with python-igraph library (*Csardi & Nepusz, 2006*). Training and validation results were tracked with Aim (*Arakelyan, 2020*) and Weights and Biases (*Biewald, 2020*) libraries.

## Complexity analysis

The training time complexity is intractable to estimate robustly as it largely depends on the number of training steps, the network size, and the implementation of the operations used within the network. In generic terms, the time complexity can be expressed as $\mathcal{O}(S(F + B))$ where $S$ is the number of training steps which can be expressed by the number of epochs times the number of minibatches within the epoch, $F$ and $B$ are the operations required for a single forward and backward pass of a minibatch respectively. $F$ and $B$ are proportional to the number of layers in the deep network $L$, and the number of nodes and edges in the graph. GCN operation is $\mathcal{O}(f \cdot (|V| + |E|))$, where $f$ is the size of the feature vector for each node. The overall time complexity would be proportional to $\mathcal{O}(S \cdot L \cdot f \cdot (|V| + |E|))$. In this approach, the training procedure converges in about 30 min and then the network can be reused for an arbitrarily constructed input graph.

The inference time complexity is proportional to the operations required for a single forward pass. For most graphs in practice, including all graphs used in this work, all the vertices in a graph can be propagated in a single minibatch, so the complexity of inference becomes $\mathcal{O}(L \cdot f \cdot (|V| + |E|))$. Further analysis of this model empirically demonstrates that $L \cdot f$ is a relatively small constant compared to other approaches and the speed of this approach outperforms others by an order of magnitude.

# EVALUATION AND RESULTS

The approach is evaluated on both real-world and synthetic graphs. Both of those are present in the benchmark provided by *Fan et al. (2019)*. The synthetic networks are generated from *powerlaw* distribution with a fixed number of edges to add ($m = 4$), and the probability of creating a triangle after adding an edge ($p = 0.05$), while the real-world graphs are taken from *AlGhamdi et al. (2017)* and represent five big graphs taken from real-world applications. The real-world graphs with their description and parameters are presented in Table 1.

The ground truth betweenness-centralities for the real-world graphs are provided by *AlGhamdi et al. (2017)*, which are computed by the parallel implementation of Brandes algorithm on a 96000-core supercomputer. The ground truth scores for the synthetic networks are provided by *Fan et al. (2019)* and are computed using the graph-tool (*Peixoto, 2014*) library.

The presented approach is compared to several baseline models. The performance of those models are adopted from the benchmark provided by *Fan et al. (2019)*:

- ABRA (*Riondato & Upfal, 2018*): Samples pairs of nodes until the desired accuracy is reached. Where the error tolerance $\lambda$ was set to 0.01 and the probability $\delta$ was set to 0.1.
- RK (*Riondato & Kornaropoulos, 2014*): The number of pairs of nodes is determined by the diameter of the network. Where the error tolerance and the probability were set similar to ABRA.
- k-BC (*Pfeffer & Carley, 2012*): Does only k steps of Brandes algorithm (*Brandes, 2001*) which was set to 20% of the diameter of the network.
- KADABRA (*Borassi & Natale, 2019*): Uses bidirectional BFS to sample the shortest paths. The variant where it computest the top-k% nodes with the highest betweenness-centrality was used. The error tolerance and probability were set to be the same as ABRA and RK.
- Node2Vec (*Grover & Leskovec, 2016*): Uses a biased random walk to aggregate information from the neighbors. The vector representations of each node were then mapped with a trained MLP to ranking scores.
- DrBC (*Fan et al., 2019*): Shallow graph convolutional network that outputs a ranking score for each node by propagating through the neighbors with a walk length of 5.

For a fair comparison, the presented model was run on a CPU machine with 80 cores and 512GB memory to match the results reported by *Fan et al. (2019)*. Please note that due to several optimizations and smaller model size, the training takes around 30 min on a single 12GB NVIDIA 1080Ti GPU machine with only 4vCPUs and 12GB RAM compared to 4.5 h reported by *Fan et al. (2019)* which used an 80-core machine with 512GB RAM, and 8 16GB Tesla V100 GPUs. For the inference, the ABCDE model does not need the 512GB memory, it only utilizes a small portion of it. Yet, the machine is used for a fair comparison. The inference is run on a CPU to be fairly compared to all the other techniques reported, yet using a GPU for inference can increase the speed substantially.

**Table 1 Summary of real-world datasets. Where |V| is the number of nodes, |E| is the number of edges, and $\overline{D}$ is the average degree of the graph. Adapted from *Fan et al. (2019)*.**

| Network | |V| | |E| | $\overline{D}$ | Diameter | Description |
|---|---|---|---|---|---|
| com-Youtube | 1,134,890 | 2,987,624 | 5.27 | 20 | A video-sharing web site that includes a social network. Nodes are users and edges are friendships |
| Amazon | 2,146,057 | 5,743,146 | 5.35 | 28 | A product network created by crawling the Amazon online store. Nodes represent products and edges link commonly co-purchased products |
| Dblp | 4,000,148 | 8,649,011 | 4.32 | 50 | An authorship network extracted from the DBLP computer science bibliography. Nodes are authors and publications. Each edge connects an author to one of his publications |
| cit-Patents | 3,764,117 | 16,511,741 | 8.77 | 26 | A citation network of U.S. patents. Nodes are patents and edges represent citations. In our experiments, we regard it as an undirected network |
| com-lj | 3,997,962 | 34,681,189 | 17.35 | 17 | A social network where nodes are LiveJournal users and edges are their friendships |

Results on real-world networks presented in Table 2 demonstrate that the ABCDE model outperforms all the other approaches for the ranking score Kendall-tau and is especially good for large graphs. For the Top-1%, Top-5% and Top-10% accuracy scores, ABCDE outperforms other approaches on some datasets, while shows close-to-top performance on others. The presented algorithm is the fastest among all the baselines and outperforms others by an order of magnitude.

Comparison of the ABCDE model with the previous GCN approach DrBC, demonstrated in Table 3, shows that the presented deep model is more accurate and can achieve better results even though it has fewer trainable parameters and requires less time to train.

The results on synthetic datasets demonstrated in Table 4 show that ABRA performs well on identifying Top-1% nodes in the graph with the highest betweenness-centrality score, even though requiring a longer time to run. On all the other metrics including Top-5%, Top-10%, and Kendall tau scores ABCDE approach outperforms all the others. ABCDE is substantially faster than others on large graphs and for the small graphs, it has comparable performance to DrBC.

It is important to note that the presented model has only around 70,000 trainable parameters and requires around 30 min to converge during training as opposed to DrBC which has around 120,000 trainable parameters and requires around 4.5 h to converge.

More GCN layers in the model enable the process to explore wider neighborhoods for each vertex in the graph during inference. *Fan et al. (2019)* used only five neighbor aggregations which limit the information aggregated especially for big graphs. We use a deeper network with more neighbor aggregations on each stage, therefore helping the network explore a wider spectrum of neighbors. That helps the network have better performance even though the structure is way simpler.

To be able to have a deep network with many graph-convolutional blocks, progressive DropEdge along with skip connections is used. Each GCN block gets only part of the graph where a certain number of edges are removed randomly. Initial layers get fewer edges, while layers closer to the final output MLP get more context of the graph which helps the model explore the graph better.

**Table 2 Top-k% accuracy, Kendall tau distance, (×0.01), and running time on large real-world networks adapted from *Fan et al. (2019)*.** It was not feasible to calculate the results marked with NA. The bold results indicate the best performance for a given metric.

| Dataset | ABRA | RK | KADABRA | Node2Vec | DrBC | ABCDE |
|---|---|---|---|---|---|---|
| Top-1% | | | | | | |
| com-youtube | **95.7** | 76.0 | 57.5 | 12.3 | 73.6 | 77.1 |
| amazon | 69.2 | 86.0 | 47.6 | 16.7 | 86.2 | **92.0** |
| Dblp | 49.7 | NA | 35.2 | 11.5 | 78.9 | **79.8** |
| cit-Patents | 37.0 | **74.4** | 23.4 | 0.04 | 48.3 | 50.2 |
| com-lj | 60.0 | 54.2* | 31.9 | 3.9 | 67.2 | **70.9** |
| Top-5% | | | | | | |
| com-youtube | **91.2** | 75.8 | 47.3 | 18.9 | 66.7 | 75.1 |
| amazon | 58.0 | 59.4 | 56.0 | 23.2 | 79.7 | **88.0** |
| Dblp | 45.5 | NA | 42.6 | 20.2 | 72.0 | **73.7** |
| cit-Patents | 42.4 | **68.2** | 25.1 | 0.29 | 57.5 | 58.3 |
| com-lj | 56.9 | NA | 39.5 | 10.35 | 72.6 | **75.7** |
| Top-10% | | | | | | |
| com-youtube | 89.5 | **100.0** | 44.6 | 23.6 | 69.5 | 77.6 |
| amazon | 60.3 | **100.0** | 56.7 | 26.6 | 76.9 | 85.6 |
| Dblp | **100.0** | NA | 50.4 | 27.7 | 72.5 | 76.3 |
| cit-Patents | 50.9 | 53.5 | 21.6 | 0.99 | 64.1 | **64.9** |
| com-lj | 63.6 | NA | 47.6 | 15.4 | 74.8 | **78.0** |
| Kendall tau | | | | | | |
| com-youtube | 56.2 | 13.9 | NA | 46.2 | 57.3 | **59.8** |
| amazon | 16.3 | 9.7 | NA | 44.7 | 69.3 | **77.7** |
| Dblp | 14.3 | NA | NA | 49.5 | 71.9 | **73.7** |
| cit-Patents | 17.3 | 15.3 | NA | 4.0 | 72.6 | **73.5** |
| com-lj | 22.8 | NA | NA | 35.1 | 71.3 | **71.8** |
| Time/s | | | | | | |
| com-youtube | 72,898.7 | 125,651.2 | 116.1 | 4,729.8 | 402.9 | **26.7** |
| amazon | 5,402.3 | 149,680.6 | 244.7 | 10,679.0 | 449.8 | **63.5** |
| Dblp | 11,591.5 | NA | 398.1 | 17,446.9 | 566.7 | **104.9** |
| cit-Patents | 10,704.6 | 252,028.5 | 568.0 | 11,729.1 | 744.1 | **163.9** |
| com-lj | 34,309.6 | NA | 612.9 | 18,253.6 | 2,274.2 | **271.0** |

# ABLATION STUDIES

To demonstrate the contribution of each part of the ABCDE approach, each part is evaluated in ablation studies. Parts of the approach are removed to demonstrate the performance changes on the real-world datasets.

From the experiments demonstrated in Table 5, it can be observed that each part's contribution differs for different graph types. ABCDE with no DropEdge outperforms the proposed approach on the com-youtube and amazon graphs which are relatively small networks. Constant DropEdge of 0.2 outperforms all the rest on the Dblp graph which is larger than com-youtube and amazon but smaller than cit-Patents and com-lj. ABCDE

**Table 3 Comparison of Top-k% accuracy, Kendall-tau, and running time on large real-world networks with the baseline DrBC model.** Results are taken from *Fan et al. (2019)*. The bold results indicate the best performance for a given metric.

| Network | DrBC Top-1% | ABCDE | DrBC Top-5% | ABCDE | DrBC Top-10% | ABCDE | DrBC Kendall-tau | ABCDE | DrBC Time/s | ABCDE |
|---|---|---|---|---|---|---|---|---|---|---|
| com-youtube | 73.6 | **77.1** | 66.7 | **75.1** | 69.5 | **77.6** | 57.3 | **59.8** | 402.9 | **26.7** |
| amazon | 86.2 | **92.0** | 79.7 | **88.0** | 76.9 | **85.6** | 69.3 | **77.7** | 449.8 | **63.5** |
| Dblp | 78.9 | **79.8** | 72.0 | **73.7** | 72.5 | **76.3** | 71.9 | **73.7** | 566.7 | **104.9** |
| cit-Patents | 48.3 | **50.2** | 57.5 | **58.3** | 64.1 | **64.9** | 72.6 | **73.5** | 744.1 | **163.9** |
| com-lj | 67.2 | **70.9** | 72.6 | **75.7** | 74.8 | **78.0** | 71.3 | **71.8** | 2274.2 | **271.0** |

**Table 4 Top-k% accuracy, Kendall tau, and execution time in seconds on synthetic graphs of different scales adapted from *Fan et al. (2019)*.** The bold results indicate the best performance for a given metric. For each scale, the mean and standard deviation over 30 tests are reported.

| Scale | ABRA | RK | k-BC | KADABRA | Node2Vec | DrBC | ABCDE |
|---|---|---|---|---|---|---|---|
| Top-1% | | | | | | | |
| 5,000 | **97.8 ± 1.5** | 96.8 ± 1.7 | 94.1 ± 0.8 | 76.2 ± 12.5 | 19.1 ± 4.8 | 96.5 ± 1.8 | 97.5 ± 1.3 |
| 10,000 | **97.2 ± 1.2** | 96.4 ± 1.3 | 93.3 ± 3.1 | 74.6 ± 16.5 | 21.2 ± 4.3 | 96.7 ± 1.2 | 96.9 ± 0.9 |
| 20,000 | **96.5 ± 1.0** | 95.5 ± 1.1 | 91.6 ± 4.0 | 74.6 ± 16.7 | 16.1 ± 3.9 | 95.6 ± 0.9 | 96.0 ± 1.2 |
| 50,000 | **94.6 ± 0.7** | 93.3 ± 0.9 | 90.1 ± 4.7 | 73.8 ± 14.9 | 9.6 ± 1.3 | 92.5 ± 1.2 | 93.6 ± 0.9 |
| 100,000 | **92.2 ± 0.8** | 91.5 ± 0.8 | 88.6 ± 4.7 | 67.0 ± 12.4 | 9.6 ± 1.3 | 90.3 ± 0.9 | 91.8 ± 0.6 |
| Top-5% | | | | | | | |
| 5,000 | 96.9 ± 0.7 | 95.6 ± 0.9 | 89.3 ± 3.9 | 68.7 ± 13.4 | 23.3 ± 3.6 | 95.9 ± 0.9 | **97.8 ± 0.7** |
| 10,000 | 95.6 ± 0.8 | 94.1 ± 0.8 | 88.4 ± 5.1 | 70.7 ± 13.8 | 20.5 ± 2.7 | 95.0 ± 0.8 | **97.0 ± 0.6** |
| 20,000 | 93.9 ± 0.8 | 92.2 ± 0.9 | 86.9 ± 6.2 | 69.1 ± 13.5 | 16.9 ± 2.0 | 93.0 ± 1.1 | **95.2 ± 0.8** |
| 50,000 | 90.1 ± 0.8 | 88.0 ± 0.8 | 84.4 ± 7.2 | 65.8 ± 11.7 | 13.8 ± 1.0 | 89.2 ± 1.1 | **92.1 ± 0.6** |
| 100,000 | 85.6 ± 1.1 | 87.6 ± 0.5 | 82.4 ± 7.5 | 57.0 ± 9.4 | 12.9 ± 1.2 | 86.2 ± 0.9 | **89.7 ± 0.5** |
| Top-10% | | | | | | | |
| 5,000 | 96.1 ± 0.7 | 94.3 ± 0.9 | 86.7 ± 4.5 | 67.2 ± 12.5 | 25.4 ± 3.4 | 94.8 ± 0.7 | **97.6 ± 0.4** |
| 10,000 | 94.1 ± 0.6 | 92.2 ± 0.9 | 86.0 ± 5.9 | 67.8 ± 13.0 | 25.4 ± 3.4 | 94.0 ± 0.9 | **96.8 ± 0.6** |
| 20,000 | 92.1 ± 0.8 | 90.6 ± 0.9 | 84.5 ± 6.8 | 66.1 ± 12.4 | 19.9 ± 1.9 | 91.9 ± 0.9 | **94.9 ± 0.5** |
| 50,000 | 87.4 ± 0.9 | 88.2 ± 0.5 | 82.1 ± 8.0 | 61.3 ± 10.4 | 18.0 ± 1.2 | 87.9 ± 1.0 | **91.7 ± 0.6** |
| 100,000 | 81.8 ± 1.5 | 87.4 ± 0.4 | 80.1 ± 8.2 | 52.4 ± 8.2 | 17.3 ± 1.3 | 85.0 ± 0.9 | **89.4 ± 0.5** |
| Kendall tau | | | | | | | |
| 5,000 | 86.6 ± 1.0 | 78.6 ± 0.6 | 66.2 ± 11.4 | NA | 11.3 ± 3.0 | 88.4 ± 0.3 | **93.7 ± 0.2** |
| 10,000 | 81.6 ± 1.2 | 72.3 ± 0.6 | 67.2 ± 13.5 | NA | 8.5 ± 2.3 | 86.8 ± 0.4 | **93.3 ± 0.1** |
| 20,000 | 76.9 ± 1.5 | 65.5 ± 1.2 | 67.1 ± 14.3 | NA | 7.5 ± 2.2 | 84.0 ± 0.5 | **92.1 ± 0.1** |
| 50,000 | 68.2 ± 1.3 | 53.3 ± 1.4 | 66.2 ± 14.1 | NA | 7.1 ± 1.8 | 80.1 ± 0.5 | **90.1 ± 0.2** |
| 100,000 | 60.3 ± 1.9 | 44.2 ± 0.2 | 64.9 ± 13.5 | NA | 7.1 ± 1.9 | 77.8 ± 0.4 | **88.4 ± 0.2** |
| Time/s | | | | | | | |
| 5,000 | 18.5 ± 3.6 | 17.1 ± 3.0 | 12.2 ± 6.3 | 0.6 ± 0.1 | 32.4 ± 3.8 | **0.3 ± 0.0** | 0.5 ± 0.0 |
| 10,000 | 29.2 ± 4.8 | 21.0 ± 3.6 | 47.2 ± 27.3 | 1.0 ± 0.2 | 73.1 ± 7.0 | **0.6 ± 0.0** | **0.6 ± 0.0** |
| 20,000 | 52.7 ± 8.1 | 43.0 ± 3.2 | 176.4 ± 105.1 | 1.6 ± 0.3 | 129.3 ± 17.6 | 1.4 ± 0.0 | **0.9 ± 0.0** |
| 50,000 | 168.3 ± 23.8 | 131.4 ± 2.0 | 935.1 ± 505.9 | 3.9 ± 1.0 | 263.2 ± 46.6 | 3.9 ± 0.2 | **2.2 ± 0.0** |
| 100,000 | 380.3 ± 63.7 | 363.4 ± 36.3 | 3,069.2 ± 1,378.5 | 7.2 ± 1.8 | 416.2 ± 37.0 | 8.2 ± 0.3 | **3.2 ± 0.0** |

**Table 5 Top-k% accuracy, and Kendall tau distance, (×0.01) on large real-world networks showing the ablation study for different parts of the ABCDE model.** The bold results indicate the best performance for a given metric.

| Dataset | No DropEdge | DropEdge = 0.2 | No skip connections | ABCDE |
|---|---|---|---|---|
| Top-1% | | | | |
| com-youtube | **78.5** | 77.8 | 66.5 | 77.1 |
| amazon | 86.2 | 91.0 | 85.3 | **92.0** |
| Dblp | 79.3 | **80.2** | 76.9 | 79.8 |
| cit-Patents | 47.4 | 47.1 | 37.6 | **50.2** |
| com-lj | 69.0 | 69.1 | 46.1 | **70.9** |
| Top-5% | | | | |
| com-youtube | **76.2** | 75.1 | 65.2 | 75.1 |
| amazon | **88.1** | 87.9 | 82.6 | 88.0 |
| Dblp | 72.3 | **74.2** | 71.5 | 73.7 |
| cit-Patents | 56.3 | 55.9 | 52.1 | **58.3** |
| com-lj | 75.4 | 75.4 | 62.8 | **75.7** |
| Top-10% | | | | |
| com-youtube | **78.1** | 77.1 | 67.5 | 77.6 |
| amazon | **86.1** | 85.4 | 77.6 | 85.6 |
| Dblp | 75.0 | **77.0** | 75.5 | 76.3 |
| cit-Patents | 63.4 | 63.0 | 60.4 | **64.9** |
| com-lj | **78.2** | 77.9 | 69.1 | 78.0 |
| Kendall tau | | | | |
| com-youtube | **59.8** | 59.3 | 56.8 | **59.8** |
| amazon | 77.3 | 77.5 | 70.9 | **77.7** |
| Dblp | 73.5 | **73.9** | **73.9** | 73.7 |
| cit-Patents | 73.2 | 72.8 | 71.1 | **73.5** |
| com-lj | 71.5 | 70.9 | 65.8 | **71.8** |

with Progressive-DropEdge and skip connections is the best for the largest two graphs, namely cit-Patents and com-lj. Removing skip connections from the model drops the performance significantly in all the cases.

As a lot of real-world graphs are very large, the final ABCDE approach is chosen to be the one leading to the best performance on the large networks.

The over-fitting behavior of the proposed approach is also studied in details. As demonstrated in the Fig. 2, the model without drop-edge over-fits faster than the models with a constant 0.2 DropEdge probability and the ABCDE model with progressive DropEdge. The ABCDE model over-fits less and has more stable validation loss compared to both the constant drop-edge models (0.2 and 0.8) and no drop-edge model. When the probability of dropping random edges from the input graph increases too much, the model starts to perform worse as demonstrated in Fig. 2. That is caused by the network structure being changed too much after the 0.8 dropout on the edges, and thus affecting the betweenness-centrality of the input network.

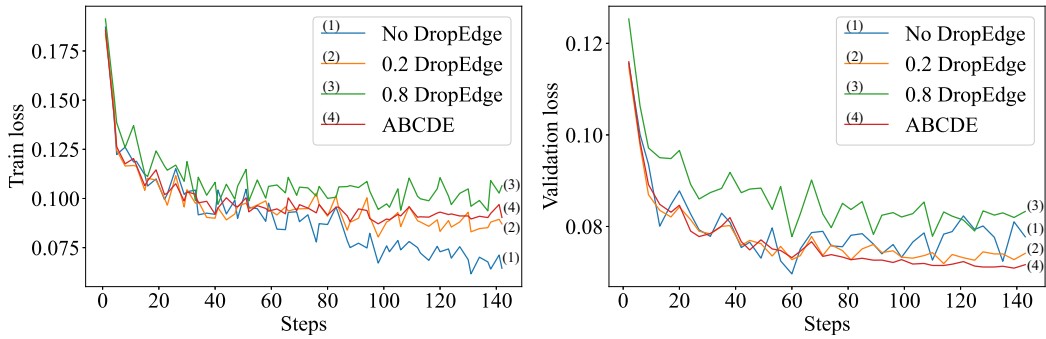

**Figure 2** The left plot represents the training losses of No DropEdge, DropEdge = 0.2 and ABCDE models; the right plot represents the validation losses of those models.

Unlike the experiments done by *Rong et al. (2020)*, there is no over-smoothing noticed in ABCDE as the model employs skip-connections for each block. That helps it avoid converging to very similar activations in deep layers.

## CONCLUSION

In this paper, a deep graph convolutional network was presented to approximate betweenness-centrality ranking scores for each node in a given graph. The author demonstrated that the number of parameters of the network can be reduced, while not compromising the predictive power of the network. The approach achieves better convergence and faster training on smaller machines compared to the previous approaches. A novel way was proposed to add regularisation to the network through progressively dropping random edges in each graph convolutional block, which was called Progressive-DropEdge. The results suggest that deep graph convolutional networks are capable of learning informative representations of graphs and can approximate the ranking score for betweenness-centrality while preserving good generalizability for real-world graphs. The time comparison demonstrates that this approach is significantly faster than alternatives.

Several future directions can be examined, including case studies on specific applications (*e.g.* urban planning, social networks), and extensions of the approach for directed and weighted graphs. One more interesting direction is to approximate other centrality measures in big networks.

### Funding

The authors received no funding for this work.

### Competing Interests

The authors declare that they have no competing interests.

## Author Contributions

- Martin Mirakyan conceived and designed the experiments, performed the experiments, analyzed the data, performed the computation work, prepared figures and/or tables, authored or reviewed drafts of the paper, and approved the final draft.

## Data Availability

The code is available on GitHub: https://github.com/MartinXPN/abcde

Reported results can be reproduced using a docker image (The docker image is available on DockerHub for Docker account holders): https://hub.docker.com/repository/docker/martin97/abcde.

By running a bash command: docker run martin97/abcde:latest.

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
