# Peer review of "ABCDE: Approximating Betweenness-Centrality ranking with progressive-DropEdge"

_PeerJ Computer Science, doi:10.7717/peerj-cs.699_

## Round 0.1 · original submission · Major Revisions

Three reviews have been received. Please provide a detailed one-to-one response. Please note you should not cite references recommended by reviewers if not appropriate. Not including such references will not affect our editorial decision.

Reviewer 1 ·

Basic reporting

The given paper uses deep graph convolutional neural network to approximate top-k nodes with highest betweenness-centrality score by estimating the rank score for each node in the graph. The proposed method obtains results which is an order of magnitude faster in inference hence requiring lesser computational resources. The literature review is structured with professional English language with proper figures and tables.

However minor concerns can be shown as follows:

1. Full form of BFS in Line 84 should be mentioned.

2. Accuracy and time can be presented in the form of a plot for easier interpretation of results

Experimental design

The research question is well designed with rigorous benchmarking across different networks and algorithms in the literature. The methods are described in detail along with the code provided for the replication of results.

Minor concerns are given as:

1. In line 154-155, authors have taken 160 graphs for training and 240 graphs for validation purpose. Generally in machine and deep learning literature, validation samples are lesser than training. Can the authors provide the rationale for this design?

2. Progressive drop edge probabilities are taken from 0.3 to 0.1. However Rong et. al (2020) have taken the probabilities from 0.5-0.8 mostly. Can the authors provide the rationale behind this design parameter?

Validity of the findings

The results provide equivalent amount of accuracy as compared to the benchmarking algorithms for top 1%, 5% and 10% scores as shown in Table 2.

However the Kendall tau coefficient is better than the other algorithms as shown in Table 2 and 3.

Additionally this algorithm also takes lesser time as compared to other benchmarking algorithms as shown in Table 2 and 3. As shown in Table 4, the algorithm scales well with the increase in the number of nodes.

However, there are certain concerns about the variant of the graph convolutional network used in this paper about the result in Table 5.

1. No DropEdge shows better results when finding top 10% nodes, however ABCDE performs much better for top 1%. Can authors explain this phenomena? Is it dependant upon the nature of the graph used for validation?.

2. How does the results change with variation in DropEdge probability as per the analysis performed in Rong et. al?

3. The authors should provide plots regarding the training and validation loss performance of the for no DropEdge condition, 0.2 DropEdge and ABCDE algorithm similar to the one presented in Rong et. al.

4. The analysis on oversmoothing, for ABCDE and DropEdge algorithms should be present comparing with different DropEdge probabilities as provided in Rong et. al.

Reviewer 2 ·

Basic reporting

1. Improve citation writing, for examples Line 26-29, 35.
2. The English language should be improved to ensure that an international audience can clearly understand your text. Some examples where the language could be improved include lines 31-32, 38-41, 55-57 etc. In general, the English language must be improved.

Experimental design

1. Line 106-107, explain in more detail the relationship of each variable in the sentence.
2. In line 112: “the sparse adjacency matrix of edges”, what is the meaning?
3. In Chapter 4: Method, sub bab 4.1 and 4.2 explain in more detail the relationship of each variable with graph notation, not just citing it. Likewise in 4.6,

Validity of the findings

Readability of the results of this paper is strongly influenced by the above improvements.

Additional comments

1. Abstract: Does not explain or does not appear to be related to the title
2. Line 25: importance node …. There is an error, node is not metric
3. In general, this paper has low readability.

Annotated reviews are not available for download in order to protect the identity of reviewers who chose to remain anonymous.

Reviewer 3 ·

Basic reporting

The authors have proposed a model for calculating approximation betweenness centrality by dropping edges progressively. The research work seems to be useful in the area of social network analysis. The authors have proposed graph convolutional networks to approximate the betweenness centrality in a large-scale complex networks.

1. The novelty of the paper is justified.
2. Motivation and application of the research work is unclear. An additional paragraph in the introduction section may be included to describe motivation.
3. What is the significance betweenness centrality in social network application? Please explain
4. Recent papers related to social network analysis especially in the area of centrality analysis, community detection should include in the paper to improve the quality. The following references are suggested to cite in the paper.
a. Behera, R. K., Naik, D., Rath, S. K., & Dharavath, R. (2019). Genetic algorithm-based community detection in large-scale social networks. Neural Computing and Applications, 1-17.
b. Kumari, A., Behera, R. K., Sahoo, K. S., Nayyar, A., Kumar Luhach, A., & Prakash Sahoo, S. (2020). Supervised link prediction using structured‐based feature extraction in social networks. Concurrency and Computation: Practice and Experience, e5839..
c. Kumar Behera, R., Kumar Rath, S., Misra, S., Damaševičius, R., & Maskeliūnas, R. (2019). Distributed centrality analysis of social network data using MapReduce. Algorithms, 12(8), 161.
d. Behera, R. K., Naik, D., Sahoo, B., & Rath, S. K. (2016, October). Centrality approach for community detection in large scale network. In Proceedings of the 9th Annual ACM India Conference (pp. 115-124).

Experimental design

1. What is the significance betweenness centrality in social network application? Please explain
2. What are the parameters you have set for GPU processing? Please explain it at the experimental setup section.
3. Is the proposed approach including any notion of distributing computing? How the work is different from the following paper.
a. Naik, D., Behera, R. K., Ramesh, D., & Rath, S. K. (2020). Map-reduce-based
centrality detection in social networks: An algorithmic approach. Arabian Journal
for Science and Engineering, 45, 10199-10222.
b. Behera, R. K., Naik, D., Ramesh, D., & Rath, S. K. (2020). Mr-ibc: Mapreduce-
based incremental betweenness centrality in large-scale complex networks.
Social Network Analysis and Mining, 10(1), 1-13

4. How your research finding is different from other works in the area of centrality analysis?
5. Which tools or package you have used to generate synthetic networks? Explain all the parameters that you have set to generate the synthetic network.

Validity of the findings

1. The authors have experimented the proposed model using several real-world and synthetic networks.
2. The results obtained is justified.
3. Conclusion supports the proposed work clearly

Additional comments

The authors have proposed a model for calculating approximation betweenness centrality by dropping edges progressively. The research work seems to be useful in the area of social network analysis. Author has proposed graph convolutional networks to approximate the betweenness centrality in large scale complex network.

1. The novelty of the paper is justified.
2. Motivation and application of the research work is unclear. An additional paragraph in the introduction section may be included to describe motivation.
3. What is the significance betweenness centrality in social network application? Please explain
4. What are the parameters you have set for GPU processing? Please explain it at the experimental setup section.
5. Is the proposed approach including any notion of distributing computing? How the work is different from the following paper.
a. Naik, D., Behera, R. K., Ramesh, D., & Rath, S. K. (2020). Map-reduce-based centrality detection
in social networks: An algorithmic approach. Arabian Journal for Science and Engineering, 45,
10199-10222.
b. Behera, R. K., Naik, D., Ramesh, D., & Rath, S. K. (2020). Mr-ibc: Mapreduce-based incremental
betweenness centrality in large-scale complex networks. Social Network Analysis and Mining, 10(1),
1-13
6. How your research finding is different from other works in the area of centrality analysis?
7. Which tools or package you have used to generate synthetic networks? Explain all the parameters that you have set to generate the synthetic network.
8. Recent paper related to social network analysis especially in the area of centrality analysis, community detection should include in the paper to improve the quality. The following references are suggested to cite in the paper.
a. Behera, R. K., Naik, D., Rath, S. K., & Dharavath, R. (2019). Genetic algorithm-based community
detection in large-scale social networks. Neural Computing and Applications, 1-17.
b. Kumari, A., Behera, R. K., Sahoo, K. S., Nayyar, A., Kumar Luhach, A., & Prakash Sahoo, S.
(2020). Supervised link prediction using structured‐based feature extraction in social network.
Concurrency and Computation: Practice and Experience, e5839..
c. Kumar Behera, R., Kumar Rath, S., Misra, S., Damaševičius, R., & Maskeliūnas, R. (2019).
Distributed centrality analysis of social network data using MapReduce. Algorithms, 12(8), 161.
d. Behera, R. K., Naik, D., Sahoo, B., & Rath, S. K. (2016, October). Centrality approach for
community detection in large scale network. In Proceedings of the 9th Annual ACM India
Conference (pp. 115-124).
9. The conclusion part supports the research work clearly.

---

## Round 0.2 · Minor Revisions

The authors should provide the training and validation curve for higher DropEdge probability, to confirm their scientific results.

Reviewer 1 ·

Basic reporting

No Comments

Experimental design

#2 - The author has claimed that the results are worse with 0.8 DropEdge probability. Can the authors provide the results for the same preferably in terms of the loss curve in Figure 2, along with the plots for No DropEdge, 0.2 probability DropEdge and ABCDE algorithm.

Validity of the findings

#1. The explanation provided is a bit weak, as the result does not seem very generalized.
One possible explanation might be due to the distribution of the betweenness centrality index of the nodes are different due to the nature of the graphs involved. Hence that causes a bit of a difference in the accuracy between top 1 and top 10%. Can the authors please investigate this?

#2.Figure 3 in Rong et. al has provided the training performance with two different values of DropEdge probability. Even if the results are worse (with higher DropEdge probability), the authors should provide the plots.

Additional comments

No Comments

---

## Round 0.3 · accepted · Accept

This version looks good. The paper can be accepted.